# A Compact Circular Rectenna for RF-Energy Harvesting at ISM Band

**DOI:** 10.3390/mi14040825

**Published:** 2023-04-08

**Authors:** Lalbabu Prashad, Harish Chandra Mohanta, Heba G. Mohamed

**Affiliations:** 1Department of Electronics and Communication Engineering, Centurion University of Technology and Management, Bhubaneswar 752050, India; 2Department of Electronics and Communication Engineering, Raghu Engineering College, Vishakhapatnam 531162, India; 3Department of Electrical Engineering, College of Engineering, Princess Nourah bint Abdulrahman University, P.O. Box 84428, Riyadh 11671, Saudi Arabia

**Keywords:** RF-energy harvesting, voltage doubler, L-section network, rectifier, wireless sensor

## Abstract

With low-power gadgets proliferating, the development of a small, effective rectenna is crucial for wirelessly energizing devices. A simple circular patch with a partial ground plane for RF-energy harvesting at ISM (2.45 GHz) band is proposed in this work. The simulated antenna resonates at 2.45 GHz with an input impedance of 50 Ω and a gain of 2.38 dBi. An L-section matching a circuit with a voltage doubler is proposed to provide excellent RF-to-DC transformation efficiency at low power input. The proposed rectenna is fabricated and the results show that the return loss and realized gain have good characteristics at the ISM band with 52% of RF-to-DC transformation efficiency, with an input of 0 dBm power. The projected rectenna is apt for power-up low sensor nodes in wireless sensor applications.

## 1. Introduction

Wireless communication has drastically advanced and subsequently increased the availability of RF signals emitted by RF sources such as digital TV transmitters, Wi-Fi routers, mobile base stations, and radio transmitters [1]. The abundant accessibility of RF energy promotes the ideation of RF-energy scavenging. For the past few years, researchers and academicians have been focused on developing green, safer, and self-sustaining technology. Thus, RF-energy harvesting is the best option to design potentially low-cost and batteryless devices. RF antenna, impedance matching circuit, and rectifier are the key modules of an RF-energy harvesting system. The antenna acts as a transducer and absorbs RF energy from the surroundings. Pairing impedance between the transducer and the rectifier is then established by matching the networks of almost all of the energy absorbed by the antenna, which must reach the rectifier without any loss, and then the rectifier changes the received RF energy to DC power [2]. To extract RF energy from the surroundings, a single-band antenna with high power density and a multiband or wideband antenna with low and medium power density are used to increase the efficiency of the rectenna. In [3], air sandwiched between FR4 substrates is used to obtain more gain at 2.45 GHz, and is designed with an open-stub rectifying network to achieve 0.46 V at 0 dBm power input. In [4], an aperture-coupled pi-shaped slot rectenna is proposed to achieve a good PCE rate of 68.83% with a 0.167 V DC output voltage at 2.45 GHz, and 49.90% with 0.236 V DC output voltage at 5 GHz. In [5], a compact wideband rectenna is designed to obtain a better conversion efficiency by 58%, and 1.6 V DC output voltage at 2.45 GHz with 0 dBm power input. A low-power multiband or broadband antenna array can gather surrounding high RF energy from multiple frequencies. However, the wideband or multiband antenna array is usually very large in size, and its complicated construction may cause coupling between antenna parts, thus reducing their efficiency [1]. A monopole antenna to harvest RF energy could be any shape, i.e., rectangular [6], fractal [7], circular [8,9,10], Vivaldi antenna [5], etc., and is a good choice to use as a receiving antenna. In [11], a 3D broadband discone antenna is proposed to harvest RF energy from cellular operating bands alongside a Dickson 2-stage voltage doubler rectifier network to charge a storage capacitor of 330 μF with 10 KΩ load resistance. Schottky HSMS 2862 diode is used as a rectifying element due to its low voltage drop, low contact capacity, and sensitivity of 35 mV/μW at 2.45 GHz. This rectenna is bulky and not recommended for portable devices. In [12], a high gain CPW fed with grounded broadband cross-mark slot rectenna is designed to absorb RF energy from 2.2 GHz to 2.6 GHz with 13 dBm input power, and 72.5% highest efficiency is observed at 2.45 GHz with 900 Ω load. The rectifier circuit is designed between the coplanar waveguide-fed gap, which causes a mismatch of impedance that reduces the efficiency from 72.5% to 50% in the broadband range. In [13], a coplanar monopole rectangular patch with two engraved U-slots is designed to operate at 2.45 GHz and to make the rectenna compact. The rectifier circuit is combined directly with the antenna, which shifts the operating frequency. The trade-off between antenna miniaturization and performance degradation is observed. The smaller antenna has a maximum gain of 0.8 dBi and the measured power conversion efficiency of 20% is observed at −20 dBm input power with a load of 4.7 KΩ. For Bluetooth/wireless local area network applications operating at 2.45 GHz, a compact novel coplanar waveguide-fed rectenna with greater efficiency is presented on the FR4 substrate. Impedance matching and bandwidth are improved by using a tuning stub approach with rectangular slots to obtain a peak gain of 5.6 dBi. The advanced design system is used to create a single-stage Cockcroft–Walton rectifier that is constructed on an FR4 substrate with an L-shaped impedance-matching network. While the measured peak conversion efficiency was found to be 68% at a 5 dBm input signal power at 2.45 GHz with an output voltage of 3.24 V, the simulated peak conversion efficiency was achieved at 75.5% [14]. In [15], a rectangular patch engraved with fractal geometry is introduced to have high bandwidth and matching properties in the frequency range from 2.15 GHz to 2.9 GHz. A simple stub-matching rectifier circuit is investigated for high conversion efficiency and impedance matching with low input power density. The measured data show that the prototype rectenna provides a high efficiency of 64% at input 0 dBm and output DC 1.5 V. The authors of [16] describe a broadband energy harvesting rectenna which has been proposed for WSN application in the frequency band ranging from 1.8 GHz to 2.6 GHz. To improve the performance of the rectifier, an optimization method has been developed, based on which a prototype rectenna is generated to cover GSM1800, UMTS, and Wi-Fi bands. The conversion efficiency of the proposed rectenna shows 15% at −20 dBm input per source and 25% when all three sources are present with a total input power density of −20 dBm. In [17], a miniaturized dual-band antenna with C- and L-shaped slots is put forward for energy harvesting at 915 MHz and for transmitting the data at 2.45 GHz under implantable conditions. A simple voltage doubler circuit with HSMS 2852-rectifying diode is placed below the antenna to reduce the overall size of the rectenna. The highest conversion efficiency of 52% is observed at 5 dBm input and a 4.3 KΩ load resistance is measured. Rectenna circuits require impedance matching. It maximizes power transmissions from the receiving antenna to the rectifier and reduces reflection losses. Impedance matching circuits are of different types, such as L-section, pi-section and T-section, of which the L-section is the simplest and most commonly used for matching networks due to its simple nature. In [18], a defective rectangular patch with a coplanar waveguide-fed rectenna is presented at 2.4 GHz. Edge slots are introduced in the ground plane near the feed to optimize and increase the bandwidth and gain 4.12 dBi. A hybrid triple rectifying network with an L-section match is attached to the antenna to obtain a conversion efficiency of 66.7% with 3.47 V output DC for 3 KΩ load at 5 dBm input. In [19], a cross-shaped simple antenna with a gain of 8.6 dbi is matched directly to the rectifier network at 2.45 GHz, eliminating the second and third harmonics; therefore, the use of a band-pass filter is not needed to make the rectenna compact and simple. The impedance matching is obtained by varying the stub length and optimizing it to obtain a high conversion efficiency of 83% with a 1.4 KΩ load at −7.7 dBm input. A microstrip resonator cell-inserted circular slot antenna is presented to suppress up to the third harmonic over a 3–8 GHz bandwidth, with a resonating frequency of 2.45 GHz. A new series-parallel rectifier [20] efficiency is increased by adding proper inductance under a low power input level. The proposed design has obtained a 70.2% conversion efficiency with a 1 KΩ load. Rectifiers with compact structures, radial stub filters [21], and stub-matching circuits are designed using ADS [22]. Furthermore, rectifier configurations include single-diode, voltage doubler, and full-wave Greinacher rectifier [23]. Rectifier performance is greatly determined by the diodes, the capacitors used in the design, and also the number of stages. Based on the RF input power, operating frequency, and load, the number of stages are optimized to obtain the maximum output DC voltage. The power conversion efficiency of the rectifier changes with the change in the number of rectifier stages and with the change in the operating frequency from low to high. A 7-stage Villard rectifier [24] made of HSMS 2850 Schottky diode is presented and investigated for its functionality at 550 MHz and 900 MHz. The properties of the rectifier are examined based on the number of stages, effects of capacitance, and stage capacitance variations. The rectified output voltage of 9.17 V and 3.78 V are observed for a 0 dBm input at 900 MHz and 550 MHz with 44.4% rectifier efficiency. It is examined that the performance of the rectifier increases up to a certain stage and then degrades. A two-stage cascade L-section rectifier [25] is designed for broadband applications from 870 MHz to 2.7 GHz. The first stage of the L-section is the high pass filter which rectifies lower band frequencies up to 1.5 GHz, and the second stage of the L-section is the inductance to rectify the frequencies beyond 1.5 GHz. The optimized rectifier conversion efficiency is 30% over the frequency range from 870 MHz to 2.5 GHz at 0 dBm input and 2 KΩ load resistance. In all the above rectenna designs, the receiving antenna models possess a degree of design complexity, and it is observed that the rectifier design of the single stage voltage doubler is widely used because it is compact and requires less components, which reduces the losses during conversion and results in greater efficiency.

Here, a simple compact circular broadband antenna that has integrated with an L-section matching network and a voltage doubler rectifying network is designed using an HSMS2852 Schottky diode for RF-energy scavenging. The rectifying antenna is proposed to harvest and change RF-energy to DC at the ISM (2.45 GHz) band which has become widespread due to the advancements in communication means. A prototype of the rectenna was fabricated and measured to validate the performance at the ISM band.

## 2. Rectenna Design

### 2.1. Monopole Receiving Antenna Design

A simple compact microstrip circular patch (CMCP) antenna layout geometry and the fabricated antenna are depicted in Figure 1. The CMCP antenna is printed on a cheap FR4 material with an $\varepsilon_{r}=4.3$, and loss tangent (tanδ) is 0.025. The substrate length and width are L_S_ and W_S_ with a thickness of 0.8 mm. A monopole circular-shaped patch of radius r is printed on the substrate and fed with a microstrip line of length and width of L_F_ and W_F_. To improve the impedance bandwidth, a reduced ground plane is placed on the back side of the substrate material with the length and width of L_G_ and W_G_. The geometrical variables of the CMCP antenna are depicted in Table 1.

The circular patch radius r is obtained by using effective radius *r_e_*
(1)$$r_{e}=\frac{8.79\times 10^{9}}{f_{r}\sqrt{\varepsilon_{r}}}$$
(2)$$r=\frac{r_{e}}{{\left(1+\frac{2h}{\pi \varepsilon_{r}r_{e}}\left[\ln\left(\frac{1.57r_{e}}{h}\right)+1.78\right]\right)}^{\frac{1}{2}}}$$

A simple circular patch antenna can be analyzed as a parallel connection of capacitance C_p_, inductance L_p_, and resistance R_p_. The partial ground plane (PGP) is analyzed as a parallel combination of capacitance C_g_, inductance L_g_, and resistance R_g_. The two resonant circuits, one for the circular patch and another one for the partial ground plane, are coupled through the mutual capacitance C_S_. Thus, the overall analogous circuit of the designed CMCP antenna is depicted in Figure 2 and simulated using ADS. The S-parameter is compared with the CST model of the antenna for validation of the equivalent circuit model in the ADS and depicted in Figure 3.

### 2.2. Design of the Rectifier

In the rectenna design, rectifying network is more important because it determines the RF-signal-to-DC transformation efficiency. A voltage doubler circuit is used as a rectifier circuit in the proposed design to obtain a good compromise among the circuit complexity and the rectifier efficiency at low input power. The designed rectifier circuit (RC) with an impedance matching network (IMN) is depicted in Figure 4. The RC with the IMN is simulated using Keysight ADS software, and co-simulator harmonic balance is used to explore the characteristics of the RC. A simple L-section IMN is connected among the RF source and the rectifier to improve the conversion efficiency by transferring maximum power from the RF source output to the rectifier input and also by reducing the components required for the circuit and thereby the miniaturization of the device. The IMN consists of a parallel-series combination of 1.4 pF capacitance and an 11.4 nH inductor is utilized to match the RF source impedance of 50.1 + 3.65 i with the 21.3 − 150.5 i rectifier impedance at 2.45 GHz. The L-section IMN is optimized using the smith chart matching technique in the ADS at 2.45 GHz.

The RC is a voltage doubler network consisting of a bypass capacitor C_1_ = 100 pF, a pair of Schottky diodes used for rectifying purposes, a shunt capacitor C_2_ = 100 pF, and an optimal load resistance of RL = 10 KΩ. In rectifying the circuit, choosing the proper diode is a critical step because it changes the received AC signal into a usable DC signal. Higher rectifying efficiency can be obtained by using a smaller built-in voltage diode. In the proposed circuit model, HSMS2852 Schottky diode is considered because of its low forward bias voltage of 150 mV and a breakdown voltage of 3.8 V. By varying the input RF power from −30 dBm to 20 dBm, the DC voltage at the output and the RF-to-DC-transformation-efficiency plots are analyzed. When RC is straightforwardly connected to the RF source, the efficiency of the doubler network is very low than with the RC connected through IMN to the RF source. This is due to improper power transfer between the RF source output and the rectifier input. The proposed rectifier is fabricated on FR4 substrate with 1.6 mm thickness to analyze its performance, as depicted in Figure 5.

## 3. Rectenna Simulation Results and Discussion

### 3.1. Monopole Antenna Simulation Results and Discussion

The proposed monopole CMCP antenna is simulated using CST software and parameters S(1,1), and Z(1,1), and parametric analysis is carried out at 2.45 GHz. Figure 6 depicts the S11 parameter of the simulated antenna. It shows that the return loss values are −28.7 dB at 2.45 GHz and −19.5 dB at 4.29 GHz. The impedance bandwidth of the prototype CMCP antenna is 2.94 GHz from 2.06–5.0 GHz, which includes ISM band (2.45 GHz), IMT services (3.5 GHz), WiMAX applications (3.3–3.8 GHz) and satellite communications (2–4 GHz).

To understand the influence of the PGP on impedance matching and antenna performance, all the variables are kept constant, and for the different values of the PGP length simulation, the results are depicted in Figure 7.

PGP length is varied from 11 mm to 14 mm and it is observed that there is a significant change in bandwidth and impedance matching. Increasing the length of the PGP from the initial value improves the impedance matching and bandwidth, and if the increasing continues, at the final value, both bandwidth and impedance matching decreases. At 12.5 mm of the PGP length, good impedance matching and wide bandwidth were observed. The wide bandwidth of 2.94 GHz from 2.06 GHz to 5.0 GHz, and the impedance of the designed CMCP antenna, which is 50 ohms at the optimal PGP length, is equal to 12.5 mm. The monopole CMCP antenna is proposed for RF-energy harvesting in which the impedance of the receiving antenna plays a crucial role in designing the IMN, whereby more energy transformation takes place from the RF source to the rectifier with a minimum power loss, improving the PCE. The impedance of the proposed antenna at PGP length of 12.5 mm is analyzed for the real and imaginary parts of the impedance of the antenna and 50.1 + 3.65 i is found from the graph depicted in Figure 8.

The simulated radiation efficiency and realized gain of the designed monopole CMCP antenna against frequency are depicted in Figure 9. The radiation efficiency of the antenna is greater than 86% in the entire wideband, and at the resonating frequency 2.45 GHz, the radiation efficiency is 95.4%. The realized gain of the antenna is 2.38 dBi at 2.45 GHz.

The polar plot of the E-plane and H-plane radiation patterns at 2.45 GHz for the CMCP antenna in co-polar and cross-polar are shown in Figure 10. In the E-plane, the cross-pole magnitude is 20 dB less than the co-pole magnitude, and similarly, for H-plane, the cross-pole magnitude is 83 dB less than the co-pole magnitude for 2.45 GHz. It is observed that at a resonating frequency of 2.45 GHz, the radiation patterns are approximately omnidirectional for co-pole and bidirectional for cross-pole, with very low magnitude. By studying the characteristics of the radiation patterns, the designed CMCP antenna radiates well in all directions with equal magnitude.

The performance of the CMCP antenna in terms of return loss and maximum gain are calibrated by using E5071C ENA series VNA, Agilent technologies after standard one port SOL between 2.0 GHz to 5.0 GHz. Figure 11 depicts the measurement setup and the comparison of simulated and calibrated S_11_ results of the CMCP antenna. From the plot, it is observed that calibrated results are more correlated with the simulated results. The resonant frequency of the calibrated results have moved slightly upward compared to the simulated kind which was generated unintentionally due to a manufacturing error, had a cable connector mismatch during calibration, and a manual soldering of the SMA connector. Furthermore, the calibrated S_11_ of the fabricated antenna covers the ISM band and bandwidth of 3 GHz from 2.0 GHz to 5.0 GHz with −21.0 dB and −27.6 dB at 2.5 GHz and 4.73 GHz, respectively.

### 3.2. Rectifier Results and Discussion

To assess the performance of the rectifying antenna, the RF–DC transformation efficiency is one of the most vital parameters, including antenna performance, power loss in the rectifying component, and impedance pairing between the RF source and the rectifier. RF AC–DC conversion efficiency is the ratio of the load power delivered to the amount of rectifier input power. The rectifier network with IMN is optimized for the resonating frequency of 2.45 GHz. The RF–DC transformation efficiency (power conversion efficiency (PCE)) can be acquired by using Equation (3).
(3)$$\eta =\frac{P_{DCout}}{P_{ACin}}\times 100$$
(4)$$P_{DCout}=\frac{{\left(V_{DCout}\right)}^{2}}{R_{L}}$$
where *P_DCout_* represents the output load DC power, *P_ACin_* represents the rectifier input power, *V_DCout_* represents the DC output load voltage, and *R_L_* represents the resistance load.

Simulated plots are obtained for the rectifier PCE, output DC voltage, and output DC power versus RF-input power sweep from −30 dBm to 20 dBm with an optimum load resistance of 10 KΩ. The plots are depicted in Figure 12.

From Figure 12a, it is found that in the absence of IMN the rectifier, efficiency is very low, i.e., 8% at 0 dBm power input, and in the presence of IMN, the rectifier efficiency is drastically improved, i.e., 60.6% at 0 dBm power input. The maximum efficiency of 63% is obtained at −2.4 dBm RF input power. The conversion efficiency slowly increases with a rise in the RF input power and reaches its peak before gradually reducing due to the nonlinear nature of the LC-IMN which was designed to operate at a resonating frequency according to the rectifying component (diode). From Figure 12b, it is noted that the output DC voltage is 2.46 V, and the output DC power is 0.6 mW at an input RF power of 0 dBm. The highest output DC voltage and power observed are 3.65 V and 1.34 mW at 10 dBm input RF power, but the rectifier efficiency is very low because the Schottky diode HSMS 2852 almost reaches its breakdown voltage. Based on the output DC power obtained at 0 dBm, it is clear that the proposed rectenna is useful to power up some wireless sensor that consumes less power than 0.6 mW, such as wearable sensor-60 µW, smoke detector-55 µW, and Wi-Fi flash memory-210 µW.

## 4. Rectenna Calibration

The measurement setup of the fabricated rectenna is considered as per reference [14], and the indoor environment is depicted in Figure 13. It consists of a traditional WLAN router considered as an RF source, which is placed 35 cm from the rectenna, and the voltmeter is connected to the output. The fabricated antenna is connected to the rectifier and it is placed in front of the NETGEAR (AC1200) WLAN router module, which has a built-in feature of changing RF signal power, thereby making the rectenna calibration simple. As we know, if we increase the distance between the RF source and the rectenna, there is deterioration in the performance of the rectified output so the separation between the RF source and the rectenna is optimized to obtain the maximum power conversion efficiency. The received power of the antenna from the RF source is calculated using Friis transmission formula, which is as follows:(5)$$P_{Rx}=G_{Tx}G_{Rx}P_{Tx}{\left(\frac{\lambda }{4\pi R}\right)}^{2}$$
where *λ* is the resonating wavelength, *G_Tx_* and *G_Rx_* are the transmitting and receiving gains of the antenna, *P_Tx_* is the power transmitted, and *R* is the separation between the RF source and the receiver.

The comparison of the measured and simulated conversion efficiency and output DC voltage and power of the rectenna is depicted in Figure 14. From Figure 14a, it can be observed that the efficiency of the proposed rectenna increases up to 0 dBm and then decreases gradually. The maximum efficiency of 52% with 2.17 V DC voltage and 0.48 mW output power is obtained at 0 dBm input power with a load resistance of 10 KΩ at 2.45 GHz. The calibrated conversion efficiency is less than simulated due to the improper soldering of the surface-mounted-device components in the rectifier network and also their nonlinear behavior at high frequencies. The higher DC voltage of 4.2 V and 1.77 mW output power is observed at 20 dBm power input. The measured and simulated results are closely correlated to one another. The comparison of the rectenna design put forward in this work with the available related models in the literature is depicted in Table 2.

## 5. Conclusions

In this work, a CMCP monopole antenna has been reported for surrounding RF-energy harvesting at ISM band. The proposed antenna is optimized to 50 Ω impedance at the operating frequency using a partial ground plane. The results show an approximate omnidirectional radiation pattern in both E-plane and H-plane with a gain of 2.38 dBi at 2.45 GHz. Moreover, a simple efficient rectifier with L-section IMN is designed and its performance is analyzed at 2.45 GHz. The rectenna is fabricated and tested at 2.45 GHz and the measured results demonstrate that the proposed rectifier network provides 2.17 V and 0.48 mW of output DC voltage and power, respectively, with a 52% rectifier efficiency at a small input RF power of 0 dBm. This describes the potential of the rectifier to collect ambient RF power and convert it into usable DC power, resulting in a fabricated prototype rectenna model that is apt for powering low sensor applications, making them sustainable, and for the model’s use in wireless sensor network applications. Future aspects that are recommended to be further explored include the design of a multiband rectenna and to evaluate the multistage rectifier so as to obtain more efficiency at multiple RF-input wavelengths, as well as to design a compact rectenna for on-body applications.

## Figures and Tables

**Figure 1 micromachines-14-00825-f001:**
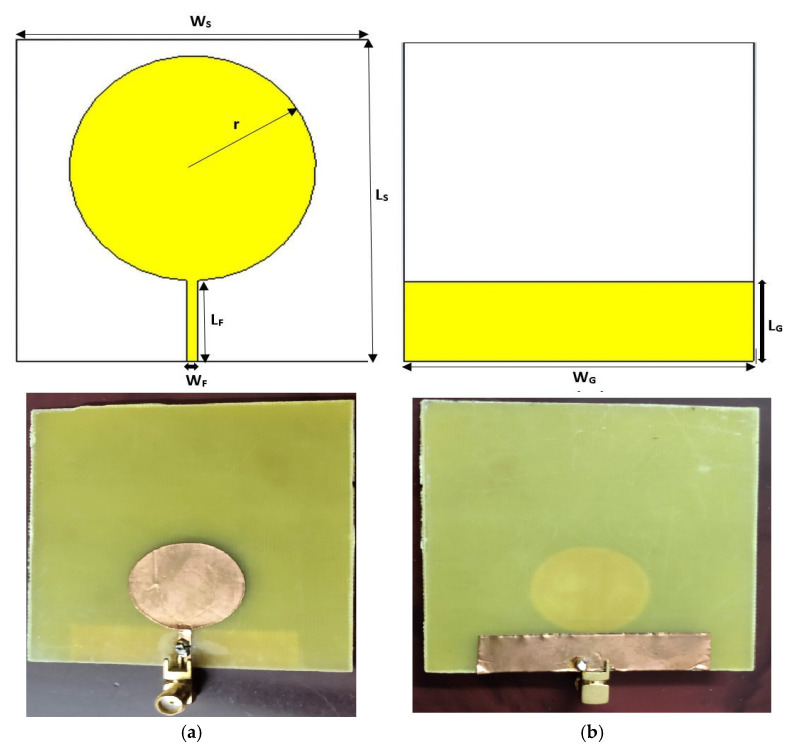
The geometry of the compact microstrip circular patch antenna and photo of the fabricated antenna: (**a**) top face; (**b**) rear face.

**Figure 2 micromachines-14-00825-f002:**
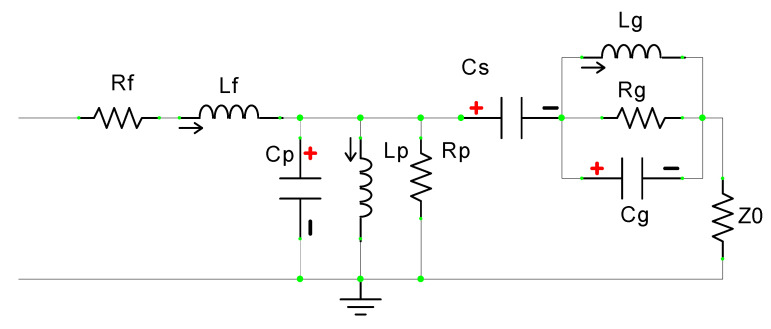
The analogous circuit of the proposed CMCP antenna.

**Figure 3 micromachines-14-00825-f003:**
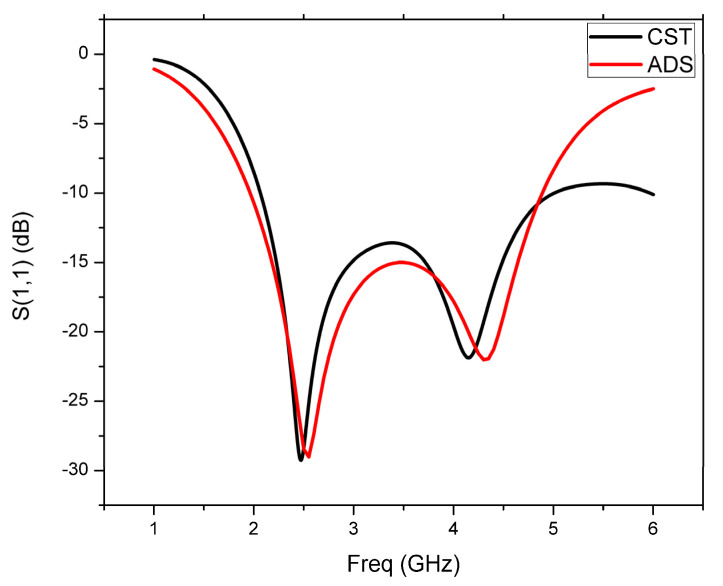
Comparison of the S11 plots acquired from the CMCP antenna and the equivalent circuit.

**Figure 4 micromachines-14-00825-f004:**
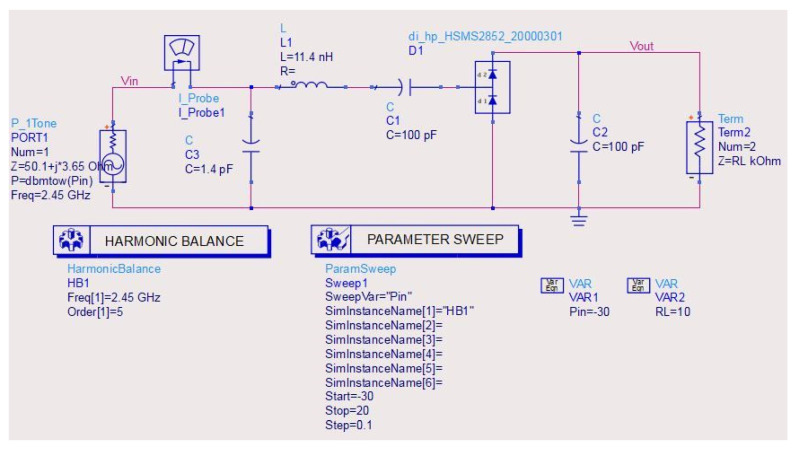
The proposed simple voltage doubler rectifier circuit with L-section IMN.

**Figure 5 micromachines-14-00825-f005:**
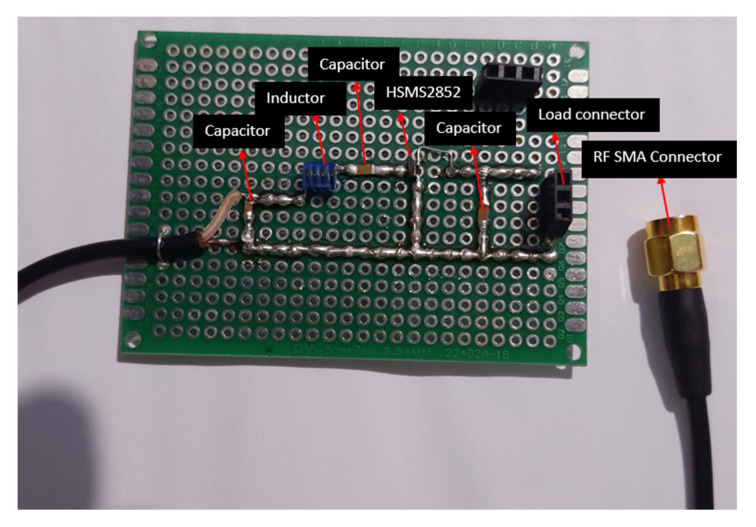
Photo of the fabricated rectifier.

**Figure 6 micromachines-14-00825-f006:**
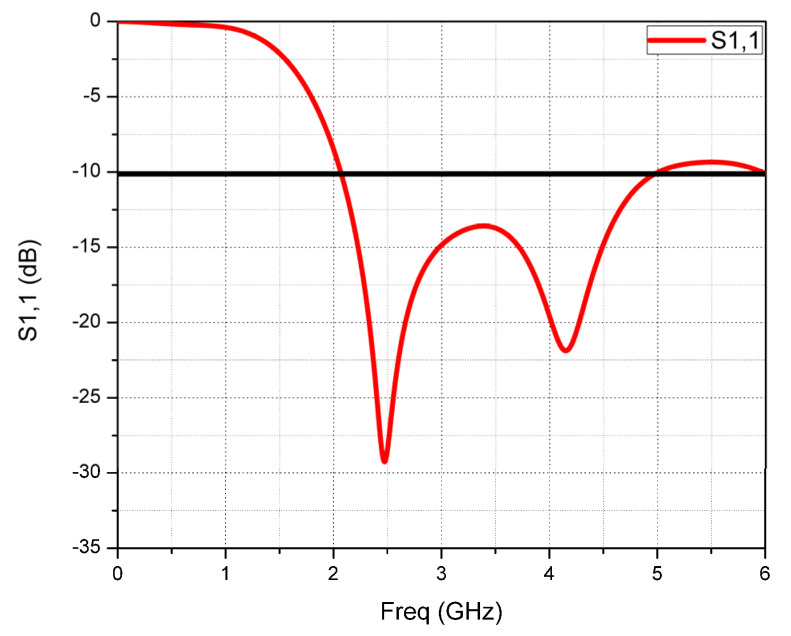
Monopole CMCP antenna S(1,1) plot.

**Figure 7 micromachines-14-00825-f007:**
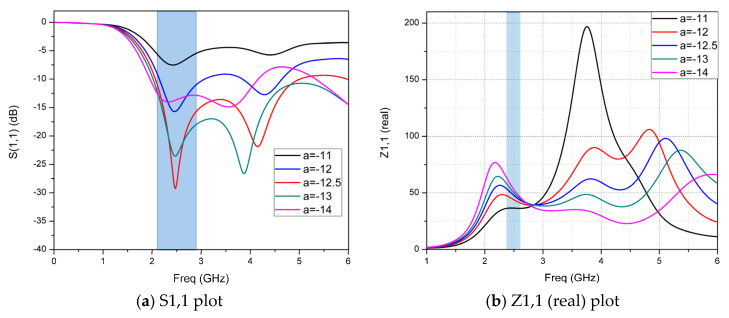
(**a**) S1,1 and (**b**) Z1,1 results for different values of the PGP length.

**Figure 8 micromachines-14-00825-f008:**
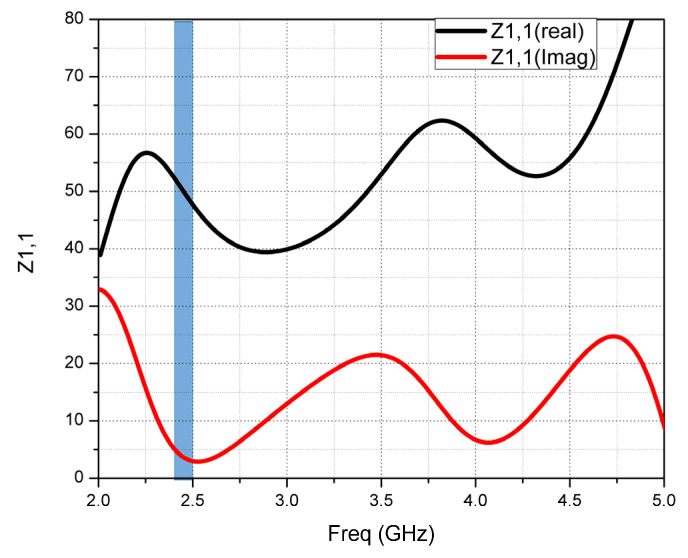
Z1,1 plot of the CMCP antenna at L_G_ = 12.5 mm.

**Figure 9 micromachines-14-00825-f009:**
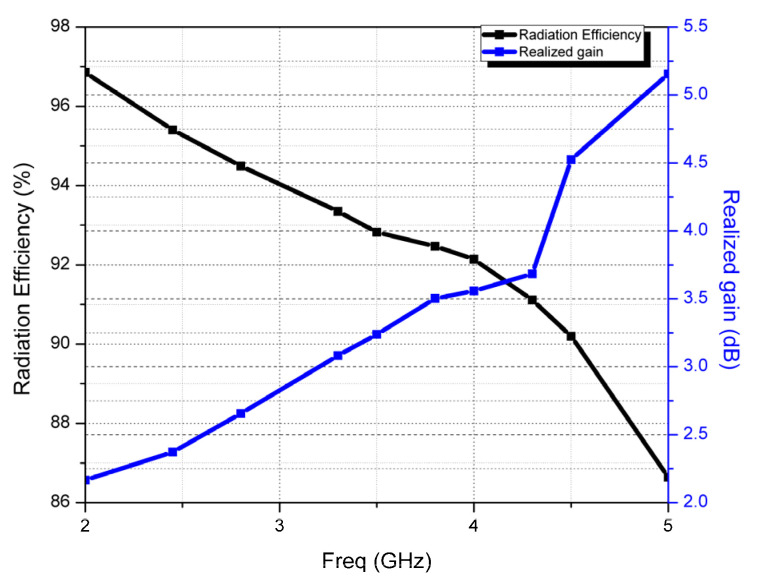
Radiation efficiency and gain plot of the CMCP antenna.

**Figure 10 micromachines-14-00825-f010:**
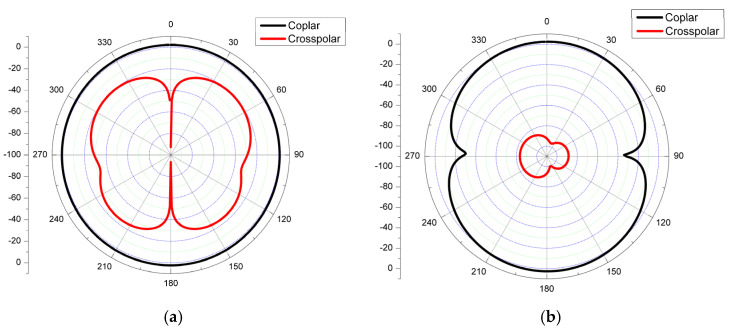
Two-dimensional polar plot radiation patterns at 2.45 GHz (**a**) E-plane (**b**) H-plane.

**Figure 11 micromachines-14-00825-f011:**
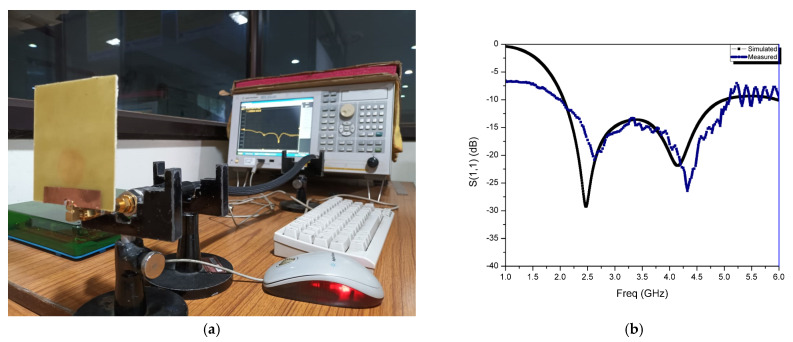
(**a**) Photo of the measurement setup for CMCP antenna (**b**) S_11_ comparison of the measured results with the simulated results for the fabricated antenna.

**Figure 12 micromachines-14-00825-f012:**
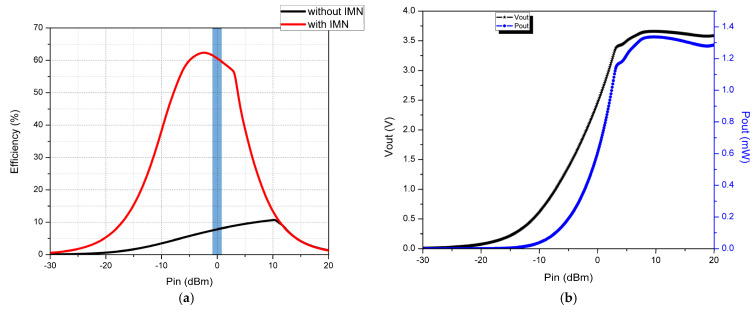
Simulated plots of the proposed rectifying network at 2.45 GHz with RL of 10 KΩ (**a**) Efficiency vs. P_in_ (**b**) V_out_, P_out_ vs. P_in_.

**Figure 13 micromachines-14-00825-f013:**
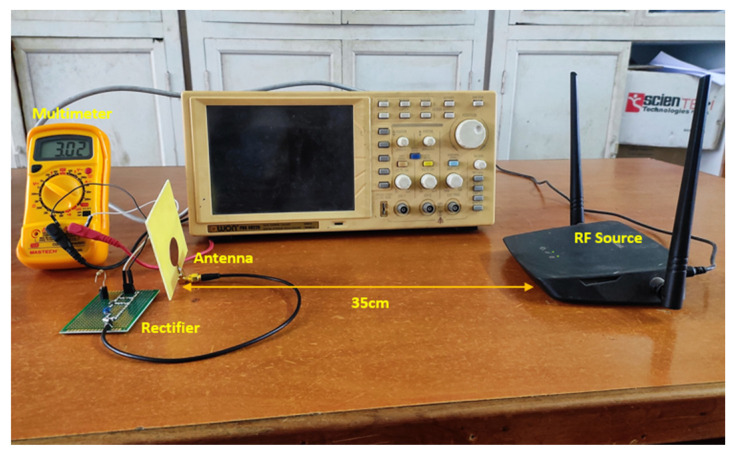
Calibration setup for the fabricated rectenna.

**Figure 14 micromachines-14-00825-f014:**
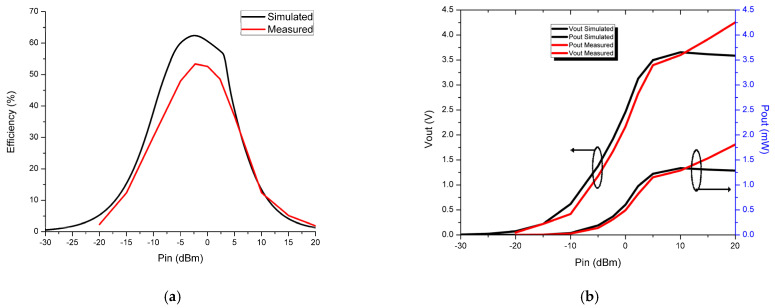
Comparison of simulated and measured (**a**) conversion efficiency (**b**) V_out_, P_out_ vs. P_in_.

**Table 1 micromachines-14-00825-t001:** The CMCP antenna’s optimal geometric dimensions.

Parameters	Values (mm)
r	17.5
W_S_	50
L_S_	50
W_F_	1.5
L_F_	12
W_G_	50
L_G_	12.5

**Table 2 micromachines-14-00825-t002:** Comparison of the rectenna model in this work with the related work available in the literature.

Reference	Antenna Size (mm^3^)	Frequency (GHz)	Rectifier Type	Input Power (dBm)	PCE (%)	Load Resistance (Ω)	Output DC Voltage (V)
[12]	135 × 93 × 1.5	2.45	Shunt and series	13	72.5	900	NA *
[13]	24.9 × 8.6 × 1.6	2.45	Voltage doubler	−20	20	4.7 K	97 m
[14]	18 × 30 × 1.6	2.45	Cockcroft–Walton	5	68	5 K	3.24
[15]	28 × 37.6 × 1.6	2.42	Voltage doubler	0	62	3.8 K	1.5
[17]	7.9 × 7.7 × 1.27	0.915	Voltage doubler	5	45	4.3 K	168.3 m
[19]	110 × 90 × 0.635	2.45	Series diode	−3.2	83	1.4 K	1.0
This work	50 × 50 × 1.6	2.45	Voltage doubler	0	52	10 K	2.17

* Not Applicable.

## Data Availability

Not Applicable.

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
