# Peer review of "A Compact Circular Rectenna for RF-Energy Harvesting at ISM Band"

_micromachines, 2023, doi:10.3390/mi14040825_

Round 1

Reviewer 1 Report

The article presents significant shortcomings both in the implementation methodology and in the state of the art of the subject. There is no comparison with other similar works and the references in the state of the art are few and poor. Has the proposed rectenna finally been implemented? If so, where is the comparison between simulation and experimental results? Please implement the required additional experiments for a comprehensive presentation of this work and expand the introduction of the article by including the following study in the citations:

[1]     Tampouratzis, M.G.; Vouyioukas, D.; Stratakis, D.; Yioultsis, T. Use Ultra-Wideband Discone Rectenna for Broadband RF Energy Harvesting Applications. Technologies 2020, 8, 21.

https://doi.org/10.3390/technologies8020021

Author Response

Thank you for the reviewer 1 comments. We have amended the manuscript as per the suggestions given by him. Please find the attachment for reference.

With Regards,

Harish

Reviewer 2 Report

The paper presents a rectenna designed for the ISM band. It consists of a circular monopole antenna, a doubling rectifier stage, and a matched filter between the two. The paper is quite well written and presents the approach well. However, the paper does not really present any originality because each of the elements taken separately (antenna, matched filter, or rectifier stage) are well known structures. Taken as a whole, this type of rectenna has already been published many times. It is therefore not interesting to publish a work based on simulations. Only a realization with tests in a real environment would be interesting for publication. On the other hand the antenna described in the paper "Dual band Wearable Antenna for ISM band Application 1Aysha Safeena A.M, 2Dr.S.Perumal Sankar, 3 Sreelekshmi published in "February 2017 IJSDR | Volume 2, Issue 2" has very similar characteristics to the proposed paper and is not in reference. Moreover, the antenna by the authors also has another potential operating frequency around 4.2GHz and is not exploited.

Author Response

Thank you for the reviewer 2 comments. We have amended the manuscript as per reviewer's suggestion. Please find the attachment for your reference.

With Regards,

Harish Chandra Mohanta

Round 2

Reviewer 1 Report

The article in its present form has been significantly improved both in the implementation methodology and in the state of the art of the subject.

The proposed rectenna has been fabricated and measured with a comparison between simulation and experimental measurements by the authors.

Reviewer 2 Report

the authors answered questions and provided measurements.